# Anisotropic attosecond charge carrier dynamics and layer decoupling in quasi-2D layered SnS$_2$

Calley N. Eads[1], Dmytro Bandak[1], Mahesh R. Neupane[2], Dennis Nordlund [iD][3] & Oliver L.A. Monti[1,4]

Strong quantum confinement effects lead to striking new physics in two-dimensional materials such as graphene or transition metal dichalcogenides. While spectroscopic fingerprints of such quantum confinement have been demonstrated widely, the consequences for carrier dynamics are at present less clear, particularly on ultrafast timescales. This is important for tailoring, probing, and understanding spin and electron dynamics in layered and two-dimensional materials even in cases where the desired bandgap engineering has been achieved. Here we show by means of core–hole clock spectroscopy that SnS$_2$ exhibits spin-dependent attosecond charge delocalization times ($\tau_{\mathrm{deloc}}$) for carriers confined within a layer, $\tau_{\mathrm{deloc}} < 400$ as, whereas interlayer charge delocalization is dynamically quenched in excess of a factor of 10, $\tau_{\mathrm{deloc}} > 2.7$ fs. These layer decoupling dynamics are a direct consequence of strongly anisotropic screening established within attoseconds, and demonstrate that important two-dimensional characteristics are also present in bulk crystals of van der Waals-layered materials, at least on ultrafast timescales.

[1] Department of Chemistry and Biochemistry, University of Arizona, 1306 East University Boulevard, Tucson, AZ 85721, USA. [2] Sensors and Electron Devices Directorate, US Army Research Laboratory, Adelphi, MD 20783, USA. [3] SLAC National Accelerator Laboratory, Stanford Synchrotron Radiation Lightsource, 2575 Sand Hill Road, MS 99, Menlo Park, CA 94025, USA. [4] Department of Physics, University of Arizona, 1118 East Fourth Street, Tucson, AZ 85721, USA. Correspondence and requests for materials should be addressed to O.L.A.M. (email: monti@u.arizona.edu)

In recent years, the family of two-dimensional (2D)-layered transition metal dichalcogenides (TMDs) has gained significant interest due to their unique layer-dependent electronic properties, including layer-dependent band structures[1], transition from indirect-to-direct bandgap[2,3], and the ability to support substantial spin[4,5] and valley[6–8] polarization. Consequently, TMDs show promise as highly efficient field-effect transistors[9,10], photovoltaics[11] and spintronic devices[12], and offer new avenues toward quantum computing. This promise hinges on a detailed understanding of the fundamental physics at play in TMDs, with particular emphasis on the spatially highly anisotropic electronic properties and the resulting consequences of quantum confinement in 2D. Indeed, exquisite band structure measurements using state-of-the-art angle-resolved photoemission (ARPES) have elucidated the electronic structure of TMDs[13–15] and opened avenues toward tailoring bandgap and electronic properties, e.g., with different interlayer twist angles[16]. Despite this emerging body of ARPES work and while the electronic structure and excitations in TMDs have been investigated widely[14,17,18], the extent to which the anisotropic electronic structure of layered materials confers 2D character already in the bulk crystals is not yet fully understood. If bulk crystals already exhibit essential aspects of 2D materials, simplification of device fabrication protocols may be anticipated, broadening their use in novel electronic devices. A number of studies have indeed suggested that layers in bulk TMDs, such as ReS$_2$[19], WSe$_2$[20], MoS$_2$[21], or in graphite[22], are sufficiently electronically decoupled to present as 2D materials, e.g., in terms of spin polarization. The observation of anisotropic screening and carrier dynamics in the time-domain would constitute a direct probe of layer decoupling in bulk crystals, but such studies are at present missing due to the extremely short timescales involved and the difficulty of spectroscopically resolving the anisotropic dynamical processes.

Here we address this open question by investigating the ultrafast carrier dynamics in the layered semiconductor SnS$_2$[23–25]. Using core–hole clock spectroscopy[26], we observe spin-dependent anisotropic charge transfer on attosecond timescales in quasi-2D bulk SnS$_2$, and show that already in bulk crystals individual layers are indeed strongly decoupled and essentially 2D. Past ultrafast spectroscopic studies have primarily focused on excitonic carrier-relaxation dynamics in bulk TMDs[27,28] with timescales of <100 fs in WSe$_2$[29], MoS$_2$[30], and SnS$_2$[31]. These measurements lack the atomic-scale specificity and time resolution to fully probe the anisotropic-correlated electron dynamics. Our study fills this need by investigating the carrier delocalization dynamics on attosecond timescales[32]: We show for the first time that van der Waals layering leads to decoupling of individual 2D layers observed dynamically on sub-fs to few-fs timescales. We demonstrate that such dynamic layer decoupling is the origin of the predominantly 2D nature and strongly anisotropic electronic properties in bulk SnS$_2$, and discuss the broader applicability to spin and electron dynamics in layered and 2D materials.

## Results

**Mapping of the conduction band of SnS$_2$.** As will be discussed in more detail below, our experimental approach is based on resonant photoemission to probe the evolution of excited states on the timescale of a core–hole decay with sub-fs time resolution. In order to properly interpret the resonant photoemission results with respect to charge-carrier dynamics, we first need to understand the character of the excited electronic states reached in the initial X-ray absorption (XA) process, in particular the excitation of electrons into the SnS$_2$ conduction band. In this study, we excite from Sn 3d levels (M-edge) to the SnS$_2$ conduction band so as to access different orbital and spin characters in the conduction

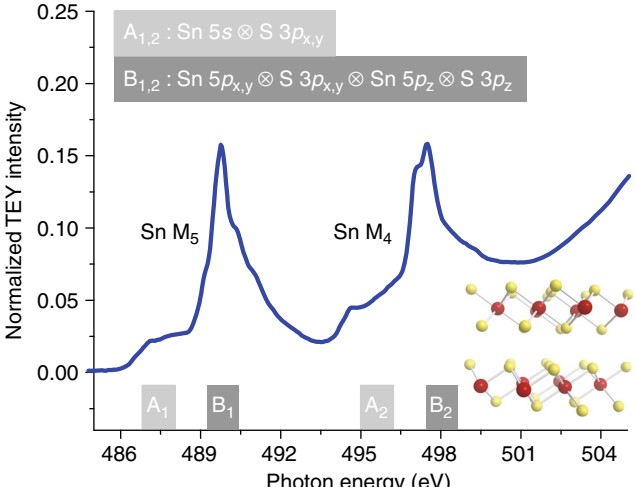

**Fig. 1** Conduction band of SnS$_2$. X-ray absorption spectra of SnS$_2$ on Sn $M_{4,5}$ edges measured in total electron yield (TEY) mode, with calculated principal orbital composition indicated. All features stem from transitions in the bulk crystal (see Supplementary Note 2 for details). Inset: Layered structure of bulk SnS$_2$ where red and gold spheres represent Sn and S atoms, respectively

band. Figure 1 shows the XA spectrum at the M-edge of SnS$_2$, separated into Sn 3d$_{3/2}$ (Sn M$_4$) and Sn 3d$_{5/2}$ (Sn M$_5$) components as a result of spin–orbit coupling. Each spin–orbit component is further split into at least two distinct features, labeled A and B. Comparison to high-level electronic structure calculations[18] allows us to assign these features to excitations of different orbital components of the conduction band (see Supplementary Note 1). The weaker features, A$_1$ and A$_2$, constitute excitations to regions near the conduction band minimum, composed predominantly of strongly hybridized Sn 5s and S 3p$_{x,y}$ (in-plane) levels. Since dipole selection rules dictate $\Delta l = \pm 1$ (with core electron orbital angular momentum $l$), the transition into Sn 5s is formally forbidden. However, we still observe a weak transition due to hybridization of Sn 5s with S 3p$_{x,y}$, resulting in orbitals/bands with dipole-allowed p-character. This transition represents excitation of primarily in-plane orbitals of both Sn and S character within a 2D SnS$_2$ sheet. In contrast, the strong XA features, B$_1$ and B$_2$, constitute dipole-allowed excitations to deeper-lying levels in the conduction band, and they are composed primarily of weakly hybridized Sn 5p$_{x,y}$, Sn 5p$_z$, S 3p$_z$, and S 3p$_{x,y}$ levels. Some of these excitations access therefore out-of-plane orbitals ultimately responsible for coupling between adjacent 2D SnS$_2$ sheets. The selective excitation in the XA process into these two groups of electronic levels provides the basis for probing the anisotropy of carrier dynamics in the van der Waals-layered material SnS$_2$. Note that surface defects do not contribute to the observed spectral features in Fig. 1 (see Supplementary Note 2 and Supplementary Fig. 1 for details).

**Strong resonant Auger enhancement channel on Sn edge**. We access the site- and element-specific ultrafast carrier dynamics in SnS$_2$ by monitoring competing decay channels following absorption of an X-ray photon from the Sn 3d levels into the conduction band region. Core-excited SnS$_2$ decays on ultrafast timescales by Auger-like processes, which are observable with resonant photoemission spectroscopy (also referred to as resonant Auger spectroscopy)[26,33]. Figure 2 illustrates the principle: In a normal Auger process (Fig. 2a, left panel), a core-excited (core-ionized) state is initially created by direct photoemission, followed by decay of the core hole and emission of a second

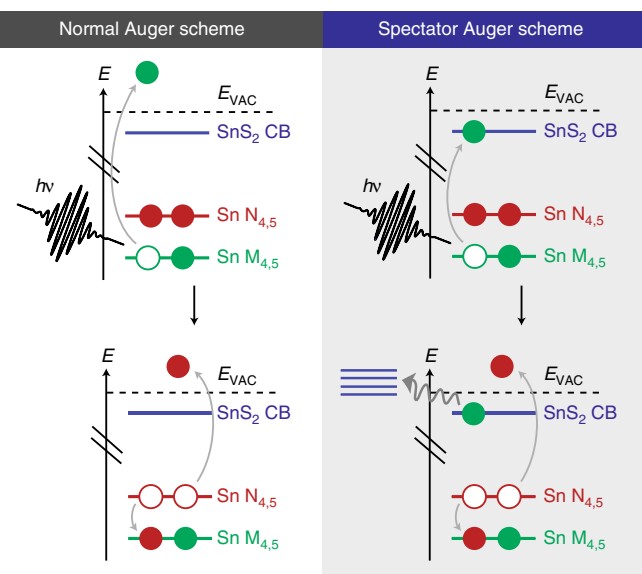

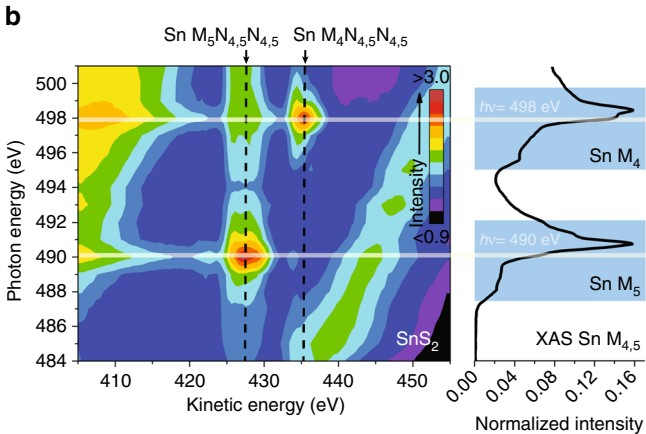

**Fig. 2** Resonant Auger scheme and intensity map. **a** Energy-level schemes of normal Auger (direct), and of spectator Auger (resonant) processes of Sn $M_{4,5}N_{4,5}N_{4,5}$ when on resonance with the conduction band of $SnS_2$. **b** Resonant photoemission contour plot of Sn $M_{4,5}N_{4,5}N_{4,5}$ compared to TEY-XAS of Sn $M_{4,5}$. Also shown are two lines (white, horizontal) for spectral cuts at constant photon energies 490 eV and 498 eV in Fig. 3

electron, and resulting in a two-hole final state. In contrast, in a spectator Auger process (Fig. 2a, right panel), resonant excitation of a core electron to a bound-excited state enables Auger-like core–hole decay during which the excited core electron acts as a spectator in the decay process. In this case, the resulting final state is a two-hole, one-electron final state. The corresponding Auger feature is thus resonantly shifted resulting in higher kinetic energy for the Auger electron feature: The spectator electron provides additional screening for the two-hole, one-electron final state, unless the excited spectator electron delocalizes on a timescale shorter than the core–hole lifetime of a few fs. In the latter case, the final state is a two-hole, zero-electron final state, and the Auger electron feature is identical to the normal Auger decay case. The competitive kinetics of electron delocalization and the known core–hole decay manifest themselves directly in the relative intensities of the resonantly shifted Auger (spectator Auger) and normal Auger features, and we use this approach to investigate the ultrafast electron dynamics in $SnS_2$.

The resonant photoemission contour plot of Auger emission intensity as a function of photon excitation energy and

photoelectron kinetic energy is shown in Fig. 2b. The total electron yield (TEY) mode absorption spectrum is obtained by integrating across all kinetic energies and shown alongside the vertical axis. Note that Auger features appear at constant values on a kinetic energy scale, indicated by vertical lines, whereas direct photoemission features disperse with photon energy, as expected for features with constant binding energy.

The intensity map shows a strong resonant enhancement of the Sn $M_{4,5}N_{4,5}N_{4,5}$ Auger features when the X-ray photon energy is scanned across the Sn $M_5$ and $M_4$ edges (dashed vertical lines). Whereas the total enhancement of all decay channels is proportional to the combined absorption cross section, the distribution between the different channels is directly related to the electronic lifetimes in the conduction band of $SnS_2$ as discussed above. We are thus able to obtain the electron dynamics from comparing the relative amplitudes of the normal and spectator Auger contributions to the Sn $M_{4,5}N_{4,5}N_{4,5}$ features in Fig. 2b. Both the normal and spectator Auger feature envelopes are analyzed in the context of atomic spectroscopy considerations of electron correlation and accessible two-hole final states (see Supplementary Notes 3 and 4 for details and Supplementary Fig. 2 for the normal Auger fit). Briefly, the normal Auger features for both Sn $M_{4,5}N_{4,5}N_{4,5}$ transitions correspond to a final state with $d^8$ atomic configuration (electronic configurations $^1S$, $^1G$, $^3P$, $^1D$, and $^3F$). We fit transitions to each of these final states by incorporating state-dependent Coulombic repulsion and a screening term in the two-hole final states, giving excellent agreement with theory[34] (see Supplementary Table 1 for quantitative results). The resulting fit parameters for the normal Auger spectra were then also used to capture the spectator Auger features after adding a spectator shift to account for the increased screening by the extra electron located in the conduction band of $SnS_2$ in resonant Auger spectroscopy.

Two representative decompositions of the Auger spectra are shown in Fig. 3a, b that are cut along the horizontal white guide lines in Fig. 2b, at photon energies of 490 eV (Sn $M_5$ resonance) and 498 eV (Sn $M_4$ resonance). At $h\nu = 490$ eV, the Auger spectrum is dominated by a resonantly shifted spectator Sn $M_5N_{4,5}N_{4,5}$ Auger contribution (spectator shift of 1.1 eV), with a minor normal Sn $M_5N_{4,5}N_{4,5}$ Auger component. There is also a small but visible excitation of normal Sn $M_4N_{4,5}N_{4,5}$. Notably, the intensity of the spectator Auger feature is approximately four times larger than that of the normal Auger feature, indicating a long-lived excited electron that screens the core–hole decay. Similarly, at $h\nu = 498$ eV, the Sn $M_4N_{4,5}N_{4,5}$ feature also has a dominant resonant Auger feature (spectator shift of 0.9 eV), but with a more significant contribution from normal Auger processes. This spectral decomposition analysis is repeated across the complete resonant photoemission map and yields the evolution of the integrated spectator and normal Auger feature intensities, as shown in Fig. 3c. The analysis shows strong resonant enhancement upon excitation of the conduction band region, and suggests the existence of long-lived carriers in some regions of the conduction band. We next use the relative magnitudes of both the normal and spectator Auger intensities to uncover the charge-carrier dynamics in $SnS_2$.

**Charge delocalization within $SnS_2$ sheets and localization between $SnS_2$ sheets**. We investigate the ultrafast charge-carrier dynamics using the core–hole lifetime as an internal clock of the excited state dynamics[26,33]. The natural lifetime of both Sn $3d_{3/2}$ and Sn $3d_{5/2}$ core–holes is identical, as predicted by theory, and verified experimentally to be $\tau_{c-h} = 1/k_{c-h} = 1.69$ fs (refs. [35–37]). The influence of Coster–Kronig decay channels associated with spin–orbit component-dependent lifetimes effects is not present

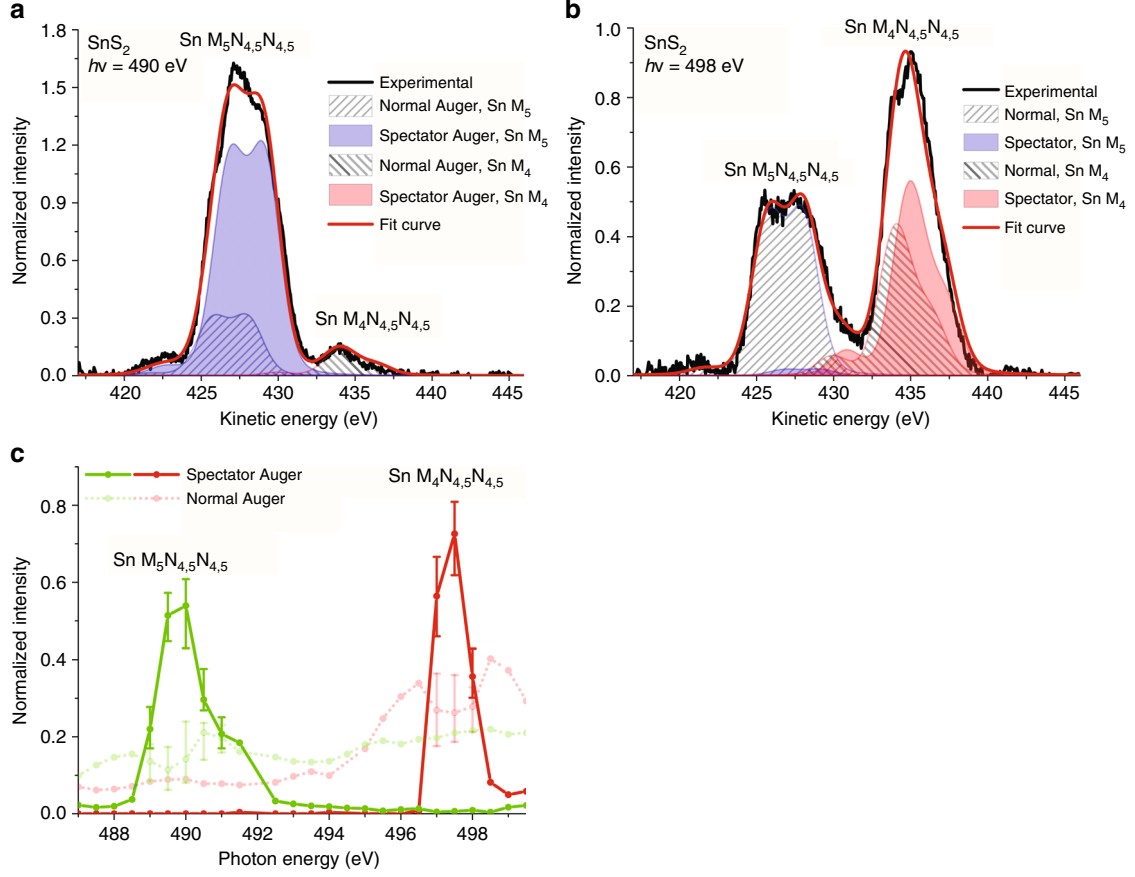

**Fig. 3** Resonant photoemission measurements at selected photon energies. Photon energy slices of the resonant photoemission spectroscopy (RPES) contour plot resonantly excited **a** on Sn $M_5N_{4,5}N_{4,5}$ at a photon energy of 490 eV and **b** on Sn $M_4N_{4,5}N_{4,5}$ at a photon energy of 498 eV, and displaying component fits of normal (dashed, gray) and spectator (solid) Auger features. **c** Integrated intensities of normal and spectator Auger components in the RPES contour plot for both Sn $M_4N_{4,5}N_{4,5}$ and Sn $M_5N_{4,5}N_{4,5}$ transitions. The error bars represent uncertainties from the fit of the spectator Auger peak areas ($\Delta\chi^2 \leq \pm10\%$)

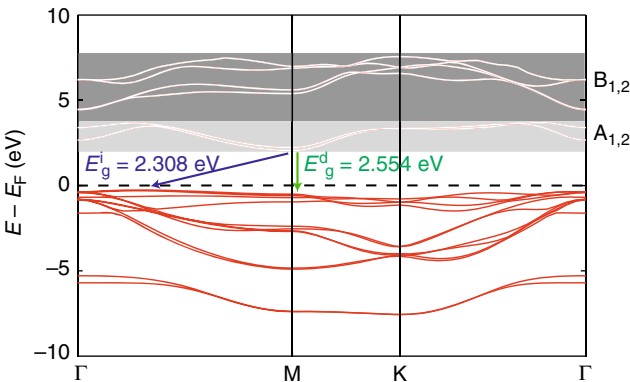

**Fig. 4** SnS$_2$ band structure. Electronic band structure of bulk SnS$_2$ with calculated indirect ($E_g^i$) and direct ($E_g^d$) bandgaps of 2.554 and 2.308 eV (see "Methods" for computational details). A$_{1,2}$ and B$_{1,2}$ indicate the regions accessed by X-ray absorption

in Sn on this edge. The experimentally accessible charge delocalization timescale is determined from the accuracy with which RPES intensities can be determined, requiring that intensities be no more than one order of magnitude apart to still enable determination of delocalization times, and hence $0.1\tau_{c-h} \leq \tau_{deloc} \leq 10\tau_{c-h}$. As a result, we expect to observe competing processes on a timescale of about 170 as to 17 fs for both

spin–orbit components. Under the assumption that core holes are strongly localized to an atomic site, charge transfer or delocalization of an excited electron away from the excited core diminishes the resonant contributions to the Auger spectrum in the spectator channel, as detailed above and in Fig. 2. The charge delocalization rate ($k_{deloc} = 1/\tau_{deloc}$) of the core-excited electron can thus be calculated relative to the competing intrinsic core–hole decay rate ($k_{c-h}$), assuming first-order rate laws for the Auger decay[26]:

$$k_{deloc} = k_{c-h} \cdot \frac{I_{NA}}{I_{SA}} \qquad (1)$$

where $I_{SA}$ and $I_{NA}$ are the intensities of the spectator and normal Auger components, respectively. The $I_{SA}/I_{NA}$ fraction varies over the investigated photon energy window of 487–499 eV, encompassing both Sn $M_{4,5}$ absorption edges and corresponding to a photon energy-dependent $k_{deloc}$. A small $I_{SA}$ fraction reflects fast-charge delocalization (high $k_{deloc}$), while a high fraction of $I_{SA}$ represents localization of the conduction band electron near the core-excited atomic site (small $k_{deloc}$). Remarkably, the energy-resolved electron dynamics vary greatly across the conduction band and even among the spin–orbit components, providing an opportunity to probe the anisotropic and spin-dependent carrier dynamics in this layered van der Waals material.

To understand the origin of the energy-dependent dynamics, we turn to electronic structure calculations. Bulk SnS$_2$ is an indirect bandgap semiconductor with a flat valence band and a maximum between the high-symmetry points Γ and M, whereas

**Table 1 Orbital compositions of SnS₂ conduction band**

|  |  | $s$ (%) |  | $p_{x,y}$ (%) |  | $p_z$ (%) |  |
|---|---|---|---|---|---|---|---|
|  |  | Sn | S | Sn | S | Sn | S |
| $\Gamma$ | $A_{1,2}$ | 48.0 | 0.26 | 0 | 47.5 | 0 | 1.55 |
|  | $B_{1,2}$ | 0.23 | 5.55 | 32.7 | 29.2 | 16.6 | 15.6 |
| M | $A_{1,2}$ | 48.1 | 2.31 | 0 | 42.6 | 0 | 7.08 |
|  | $B_{1,2}$ | 0 | 11.6 | 13.4 | 17.6 | 31.1 | 26.3 |

Normalized orbital compositions (in %) of regions $A_{1,2}$ and $B_{1,2}$ at two representative high-symmetry points in the Brillouin zone. Dominant orbitals contributing include $s$, $p_x$, $p_y$, and $p_z$ for both Sn and S atoms at $\Gamma$ and M. Sn $d$ orbitals contribute less than 3% of the total orbital contributions

the conduction band minima are at $\Gamma$ and M (see Fig. 4). The RPES results are independent of electron momentum, and we concentrate on these two representative high-symmetry points, $\Gamma$ and M, to understand the energy-dependent electron dynamics. This is further justified by the relatively minor dispersion in both the valence and conduction bands, and the fact that all our conclusions are also valid at K (see Supplementary Note 5). The full band structure, including $\Gamma$ to A, is shown in the Supplementary Note 5 (Supplementary Fig. 3).

Orbital contributions were calculated for both Sn and S at select energies and momenta in the conduction band region, and integrated over energy ranges corresponding to $A_{1,2}$ and $B_{1,2}$ identified in XAS. Both regions are dominated by Sn and S $s$ and $p$ orbitals, and contributions from Sn $d_{x,y}$, $d_{y,z}$, $d_{z^2}$, $d_{x,z}$, and $d_{x^2-y^2}$ can be neglected. A summary of the relative contributions at $\Gamma$ and M is shown in Table 1. Since RPES cannot distinguish between different points in the Brillouin zone, small variations in the overall similar orbital composition at both $\Gamma$ and M should be averaged (refer to Supplementary Table 2 for orbital composition at K). Regions $B_{1,2}$ differ markedly from $A_{1,2}$ by containing significant out-of-plane orbital character from Sn and S $p_z$ contributions and by lacking Sn 5$s$ character. As reflected in the weak spectral intensity, excitation of regions $A_{1,2}$ is by dipole selection rules only allowed due to strong hybridization with S 3$p_{x,y}$ orbitals. $A_{1,2}$ is thus primarily composed of strongly hybridized in-plane molecular orbitals. $B_{1,2}$ contains primarily Sn and S $p_{x,y}$ and $p_z$ character. Of these, the Sn and S $p_z$ orbitals are more strongly hybridized due to the atomic arrangement in the unit cell (see insert Fig. 1), and $B_{1,2}$ contains significant out-of-plane character. As a consequence, the extracted delocalization times in regions $A_{1,2}$ reflect predominantly in-plane intralayer electron dynamics, while regions $B_{1,2}$ report on out-of-plane interlayer processes. The atomic specificity of resonant photo-emission together with an analysis of the orbital composition in the conduction band of SnS₂ reveals thus dynamics of anisotropic charge flow in SnS₂.

From Eq. (1) and the energy-dependent normal and spectator Auger intensities in Fig. 3c, we calculate the energy-dependent electron delocalization times for different regions in the conduction band. The low-resonant enhancement in regions $A_{1,2}$ corresponds to highly delocalized charge carriers with ultrafast carrier delocalization times of $\tau_{\text{deloc},A_1} = \frac{1}{k_{\text{deloc},A_1}} = 376^{+100}_{-100}$ as and $\tau_{\text{deloc},A_2} = \frac{1}{k_{\text{deloc},A_2}} < 169$ as, respectively, averaged over the indicated regions in Fig. 5a. In fact, the finite spectral signal-to-noise ratio in region $A_2$ restricts the experimentally accessible temporal resolution of carrier delocalization to an upper bound. Since regions $A_{1,2}$ are principally composed of hybridized in-plane Sn 5$s$ and S 3$p_{x,y}$ orbitals at both $\Gamma$ and M and near the conduction band minimum, our results demonstrate attosecond intralayer delocalization dynamics as a consequence of strong hybridization within a single SnS₂ layer (Fig. 5b).

In contrast, charge delocalization times in regions $B_{1,2}$ are larger by at least an order of magnitude, i.e., $\tau_{\text{deloc},B_1} = \frac{1}{k_{\text{deloc},B_1}} = 3.80^{+1.42}_{-0.99}$ fs and $\tau_{\text{deloc},B_2} = \frac{1}{k_{\text{deloc},B_2}} = 2.68^{+0.83}_{-0.67}$ fs. This much-enhanced electron localization can be attributed to the difference in orbital composition between regions A and B: While Sn 5$p_{x,y}$ and S 3$p_{x,y}$ orbitals contribute at both $\Gamma$ and M to this part of the conduction band too, only regions $B_{1,2}$ have significant S 3$p_z$ and Sn 5$p_z$ orbital character. These out-of-plane orbitals are ultimately responsible for interlayer coupling and hence govern the delocalization between SnS₂ layers. Symmetry considerations derived from the unit cell of 4H-SnS₂ with symmetry group $P3m1$ validate our orbital hybridization schemes in regions $A_{1,2}$ and $B_{1,2}$ (see Supplementary Note 1). The comparatively long carrier lifetimes in these out-of-plane orbitals, only found in regions $B_{1,2}$ deeper in the conduction band, are clear evidence for weak interlayer coupling. Taken together, the strongly anisotropic nature of the electron dynamics in the conduction band region unambiguously indicate a strongly dynamical 2D nature already of bulk SnS₂.

A comparison of these delocalization times with a simple estimate of hopping times obtained from the density functional theory (DFT) band structure calculations (Fig. 4 and Supplementary Fig. 3, $\tau_{\text{hop}} = \hbar/W$ with $W$ the bandwidth) shows that such estimates (~300 as intralayer hopping and ≫1 fs interlayer hopping) are compatible with our experimentally determined delocalization times. Note, however, that hopping times and delocalization times are not expected to be identical due to the presence of the core hole in the RPES measurements and possible difficulties of DFT calculations to capture bandwidths correctly.

**Discussion**

Near the conduction band minimum, regions $A_{1,2}$ take advantage of strongly hybridized in-plane orbitals that lead to strong intralayer coupling. As a result, charge delocalization in SnS₂ takes place on timescales of less than 400 as within a single layer. Deeper in the conduction band, regions $B_{1,2}$ contain both in-plane and out-of-plane hybridized orbitals, and the electron dynamics access interlayer processes that occur on timescales at least one order of magnitude slower due to weak interlayer coupling. The underlying reason for this difference is the strongly anisotropic dynamic screening of charges in SnS₂, as already reported on long timescales from static spectroscopy, e.g., of WS₂[17] and MoSe₂[38]. The consequences are much more strongly screened Coulomb interactions and hence facile carrier delocalization within a layer. Coulombic interactions across layers are less screened and significantly longer range, bringing about slow interlayer charge transfer dynamics. This is also fully consistent with the band structure, which shows negligible dispersion along the $\Gamma$ to A direction (see Supplementary Note 5).

Our measurements constitute the first observation of anisotropic carrier dynamics in van der Waals-layered materials, complementing efforts to tailor and understand the band structure of these materials, e.g., by ARPES[13–16]. While the extent to which layer decoupling varies across this class of materials is at present unknown, we anticipate that our approach enables investigations into variations in interlayer coupling in response to stacking symmetry, twist angle and formation of heterostructures, complementing steady-state spectroscopies such as ARPES and photoluminescence[39]. Our findings further strongly suggest the possibility to create long-lived interlayer excitons[40] in few layer and even in bulk crystals, since interlayer delocalization between neighboring SnS₂ sheets is at least a factor of 10 slower than intralayer delocalization. This process may be quite fast in the presence of an energetic gradient, as indicated by a recent report of interlayer charge transfer times of <50 fs in MoS₂/WS₂

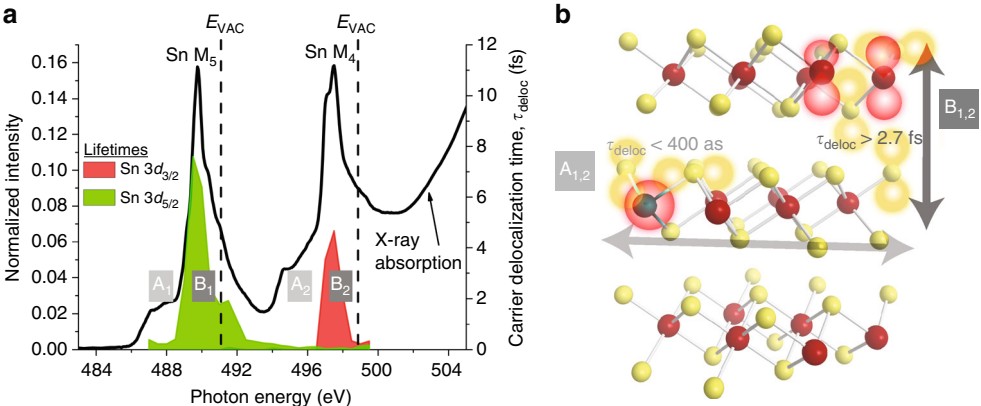

**Fig. 5** Carrier delocalization dynamics in $SnS_2$. **a** Carrier delocalization times extracted from resonant photoemission spectroscopy, superimposed with XAS. Regions $A_{1,2}$ correspond to highly delocalized conduction band electrons, while electrons excited to regions $B_{1,2}$, deeper in the conduction band are strongly localized. **b** Cartoon of strong intralayer coupling leading to delocalization times less than 400 as, and weak interlayer coupling resulting in delocalization times greater than 2.7 fs

heterostructures[41]. Note that this timescale is fully consistent with our measured interlayer delocalization times.

As a direct consequence of the anisotropic electron dynamics demonstrated here, different layers in bulk $SnS_2$ are decoupled. Such layer decoupling observed dynamically provides real-time control over ultrafast electron flow even in bulk $SnS_2$ in the device-relevant conduction band valleys. Indeed, while residual interlayer interactions exist, our data show that on ultrafast timescales bulk $SnS_2$ acts primarily as a 2D material composed of single $SnS_2$ sheets due to the highly delocalized intralayer charge dissipation and strong Coulombic localization across multiple layers. We suggest that these findings are more broadly applicable to other layered van der Waals materials such as $MoS_2$. The existence of layer decoupling, directly demonstrated here on short timescales, may thus enable harnessing some of the unique properties of 2D materials in more readily available bulk materials, facilitating integration into new device platforms.

Interestingly, our resonant photoemission data also point to the existence of spin-dependent dynamics: The two excitation edges Sn $M_5$ and $M_4$ observed in XAS differ only in the spin–orbit character ($J = 5/2$ and 3/2, respectively). Carrier lifetimes differ by approximately a factor two, offering an opportunity for dynamical spin filtering on ultrafast timescales in $SnS_2$[42]. Layer decoupling observed via dynamics on ultrafast timescales may therefore be useful not only for controlling electron dynamics, but also more generally for spin and quasiparticle currents in quasi-2D-layered materials.

## Methods

**$SnS_2$ material synthesis and sample preparation.** As reported elsewhere, the $SnS_2$ single crystals of 4H polytype were grown using the Bridgman method and n-doped with chlorine to a carrier concentration of $2.3(1) \times 10^{17}$ cm$^{-3}$ [43]. The $SnS_2$ substrate was cleaved via mechanical exfoliation before introducing the sample to the ultrahigh vacuum preparation chamber. This was followed by pump down, baking for 12 h, and annealing at ~200 °C for 12 h. XPS measurements of Sn $3d_{3/2}$ and $3d_{5/2}$ features revealed no reduction of $Sn^{4+}$ to $Sn^{2+}$.

**Synchrotron experiments.** All spectra were obtained at the SLAC National Accelerator Laboratory facility Stanford Synchrotron Radiation Lightsource (SSRL) on beamline 10–1. The $SnS_2$ single crystals were mounted on a Mo foil using tungsten wires and annealed using a UHV button heater, followed by XPS analysis before starting the RPES measurements. The RPE spectra were acquired using a double-pass cylindrical mirror analyzer, mounted in the plane of the surface normal perpendicular to the incoming radiation. The incoming synchrotron radiation is linearly polarized with the electric field vector perpendicular to surface normal. XA measurements were performed in both TEY mode, using the drain current as well as channeltron detector current, and total fluorescence yield mode using a Si photodiode (IRD AXUV100) to differentiate between surface and bulk

features, respectively. Multiple X-ray incidence angles (20°, 55°, and 90°) were measured for each absorption edge to establish angular dependence and ensure surface and bulk integrity. All spectra were calibrated against the incident X-ray photon flux measured on a gold grid installed upstream of the analysis chamber. RPES was collected at a pass energy of 25 eV in photemission mode, corresponding to 0.4 eV resolution, and the slits of the beamline spherical grating monochromator were set to approximately match this resolution for a total resolution of about 0.5 eV. The base vacuum was kept below $1 \times 10^{-8}$ Torr during the measurements, and sample integrity was checked before and after each RPES run using XPS. Photon energy calibration was achieved by fixing S $2p_{1/2}$ and S $2p_{3/2}$ at 162.5 and 163.6 eV binding energies, respectively, and by XA scans conducted before and after each RPES data set.

**Spectral corrections.** A linear background subtraction followed by an integrated background subtraction was performed on all XP and XA spectra. All spectral intensities were normalized with respect to X-ray photon flux ($I_0$) of the incoming synchrotron radiation calibrated on a gold grid upstream from the sample. Reference scans of S $2p_{1/2}$ and S $2p_{3/2}$ were measured after each RPES and XA scan for energy calibration and global vacuum-level corrections.

**Electronic structure calculations.** First-principle DFT was performed using the projector-augmented wave method[44] as implemented in the VASP package[45]. In order to account for dispersion interactions during structure optimization and electronic structure calculations, an empirical dispersion correction method proposed by Grimme et al. was used[46]. The atomic coordinates were optimized using both the non-local-correlated Perdew-Burke-Ernzerhof (PBE) functional and a hybrid functional (HSE)[47]. For the HSE calculations, Hartree–Fock screening and exchange parameters, with a PBE correlation, were set to 0.2 Å$^{-1}$ and 25%, respectively, known as the HSE06 flavor of the available hybrid functionals within the VASP package. A zone-centered Monkhorst–Pack scheme[48] was adopted to integrate over the Brillouin zone with a k-mesh of $8 \times 8 \times 4$ for the $SnS_2$ structure, and a plane-wave basis kinetic energy cutoff of 300 eV was used. To represent the crystalline solid, periodic boundary conditions in all three spatial dimensions were applied. The total energies and Hellmann–Feynman forces on the atoms were converged to $10^{-6}$ eV and 50 meV Å$^{-1}$, respectively. Space and momentum-projected DOS calculations and band structure calculations were performed by integrating over the Brillouin zone with a $16 \times 16 \times 8$ k-mesh using the tetrahedron method with Bloch corrections[44]. Spin–orbit coupling was included during density of states (DOS) and band structure calculations.

**Data availability.** The data that support the findings of this study are available from the corresponding author on request.

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

## Acknowledgements

This work was supported by the National Science Foundation, award numbers CHE 1213243 and CHE 1565497. Use of the Stanford Synchrotron Radiation Lightsource, SLAC National Accelerator Laboratory, is supported by the U.S. Department of Energy, Office of Science, Office of Basic Energy Sciences under Contract No. DE-AC02-76SF00515. M.R.N. also acknowledges the support by grants of computer time from the DOD High Performance Computing Modernization Program at the U.S. Air Force Research Laboratory and U.S. Army Engineer Research and Development Center DOD Supercomputing Resource Centers. The authors thank Bruce Parkinson for providing the bulk SnS$_2$ crystals.

## Author contributions

The experiments were planned and supervised by O.L.A.M. Measurements were performed by C.N.E., D.B. and D.N. M.R.N. performed the electronic structure calculations. C.N.E. analyzed the data. C.N.E. and O.L.A.M. wrote the manuscript in discussion with D.N., D.B. and M.R.N.

## Additional information

**Competing interests:** The authors declare no competing financial interests.

7