## [Peer Review File · Nature Communications]

Reviewers' comments:

Reviewer #1 (Remarks to the Author):

The article reports on core-hole clock spectroscopy measurements on the layered material SnS₂. The authors measure different delocalization time for carriers that are confined within the plane and out of plane. To my knowledge, this is the first time this technique has been applied on a quasi-2D material. In general, the paper is clearly written, the results of experiments are presented in an understandable manner and I have only few comments, which I present below. The Supplemental Material also provides some valuable information regarding the details of the research. However I am not convinced that the paper contains enough new physics to warrant the publication on this journal. In particular, although these results are novel, I don't think they have a significant impact for the physics of 2D or layered materials. The main conclusion of this work i.e. the fact that the delocalization time is shorter for electrons within the layer and longer for electrons in out of plane orbitals, is somehow expected because of the strong anisotropy between in plane and out of plane physical properties of the layered materials. Therefore, I find the paper more suitable for a more specialized journal.

My detailed comments:

- Normal and spectator Auger features in fig 3: The authors measure the Auger spectra far from the resonance at a photo energy of 520 eV in order to determine the normal Auger component and they use it in order to disentangle normal and spectator component. In my opinion this procedure is not well explained in the article. I suggest to the authors to move the part "Normal and spectator Auger fits" in the main manuscript. This would make the article more clear for people who are not in the field.
- Time scale determination: The authors "expect to observe competing processes on a time scale of about 170 as to 17 fs". It is not clear to me how do they estimate this interval.
- In the last part of the text, the authors talk about interlayer and intralayer hopping time. Is the hopping time equivalent to the delocalization time estimated by core-hole clock spectroscopy? In the band picture the carrier can hop from one atom to the neighbor one in a characteristic time (hopping time) $t = \hbar/W$ where W is the width of the conduction band. What is width W of the conduction band estimated by DFT calculations reported in Fig 4? Is the estimated value of W compatible with the delocalization time measured by the experiments?
- Inter and intralayer τ delocalization: Can the authors compare the values for intra and interlayer delocalization in SnS₂ with the delocalization time theoretically estimated or experimentally measured on other semiconductors or layered materials such TMDs and graphene? Is the carrier delocalization mechanism measured in SnS₂ particularly fast with respect to other materials?

Reviewer #2 (Remarks to the Author):

In this work the authors probe the excited state carrier dynamics of SnS₂ using resonant Auger spectroscopy. It is a well written paper with findings which may have wider implications on vdW layered materials. Of the major findings are that the timescale of charge delocalization in-plane is orders of magnitude larger than out of plane. Hence, even the bulk behaves somewhat as isolated 2D conductive sheets. This deduced anisotropy relies on the identification of $A_{\rightarrow 1,2\rightarrow}$ and $B_{1,2}$ conduction bands as being associated with intralayer and interlayer charge dynamics, respectively. My major concerns are listed below:

- (1) It is not clear why the hole delocalization time for $B_{1,2}$ is so long. While it is argued that the presence of p_z character means this is associated with interlayer coupling, it does not seem obvious that it is exclusively (or even predominately) interlayer in nature. Looking from the orbital

perspective, table I shows significant (30~60%) $p_{x,y}$ character. Presumably, this is also intralayer in nature and so while one may not expect the electron to delocalize quickly in z , why does it not delocalize in-plane? Could this instead be tied to a lack of any significant S_n (S) character in the band?

(2) Along a similar line, the dispersion of the bandstructure in the in-layer vs out-of-layer directions may be more convincing than relying on the orbital character. The bandstructure shown seems to be only for in-plane high symmetry points (for a hexagonal supercell?). It would be nice if we could also see Γ -A- Γ , so we could see how the effective mass of A and B in the parallel and perpendicular directions compare.

(3) The manuscript states that a "small I_{SA} fraction reflects fast charge delocalization (high k_{deloc})", and vice versa. This seems to directly contradict equation (1), $k_{deloc} = k_{(c-h)} I_{SA}/I_{NA}$. From my understanding, the localized electron screens the core-hole more effectively, so eqn. (1) seems incorrect. Should it instead be I_{NA}/I_{SA} ?

Reviewer #1

1. However I am not convinced that the paper contains enough new physics to warrant the publication on this journal. In particular, although these results are novel, I don't think they have a significant impact for the physics of 2D or layered materials. The main conclusion of this work i.e. the fact that the delocalization time is shorter for electrons within the layer and longer for electrons in out of plane orbitals, is somehow expected because of the strong anisotropy between in plane and out of plane physical properties of the layered materials.

We thank the reviewer for their description of how our work fits into a broader context. However, we respectfully disagree with the reviewer's statement regarding the novelty and impact of our results. We demonstrate for the first time that *layered materials are inherently 2D on short timescales even in their bulk form*, shown dynamically using a site-selective spectroscopic technique providing attosecond time resolution. Anisotropic ultrafast dynamics on such short time-scales requires atomic specificity, and standard ultrafast spectroscopies are unable to access the necessary dynamic regime. Instead, core-hole clock spectroscopy is uniquely suited for this purpose, as shown for the first time in our investigations.

To what extent bulk layered materials exhibit a 2D nature has been a question of significant interest in the field of 2D materials. A number of recent high impact studies have attempted to shed light on this question: Tongay *et al.*, *Nat. Commun.* **5**, 3252 (2014)¹ suggests pseudo-monolayer behavior in bulk ReS₂ using optical and vibrational spectroscopies; Riley *et al.*, *Nat. Phys.* **10**, 835-839 (2014)² found high spin polarization in bulk WSe₂ that was thought only to be present in monolayer WSe₂; Gehlmann *et al.*, *Sci. Rep.* **6**, 26197 (2016)³ found a similar result in bulk MoS₂; and Reed *et al.*, *Science* **330**, 805-808 (2010)⁴ measured the fine-structure constant in graphite assuming graphite consists of free-standing graphene sheets. Each of these studies *inferred* a 2D character in the bulk layered materials, suggesting that already bulk crystals exhibit some of the

¹ Tongay *et al.*, Monolayer behaviour in bulk ReS₂ due to electronic and vibrational decoupling, *Nat. Commun.* **5**, 3252 (2014)

² Riley *et al.*, Direct observation of spin-polarized bulk bands in an inversion-symmetric semiconductor, *Nat. Phys.* **10**, 835 (2014)

³ Gehlmann *et al.*, Quasi 2D electronic states with high spin-polarization in centrosymmetric MoS₂ bulk crystals, *Sci. Rep.* **6**, 26197 (2016)

⁴ Reed *et al.*, The effective fine-structure constant of freestanding graphene measured in graphite, *Science* **330**, 805 (2010)

important and surprising properties usually associated with the 2D limit. A more direct proof of layer decoupling requires however an experimental approach that can probe the anisotropic carrier dynamics, as demonstrated in our work.

Finally, we would like to emphasize that beyond enhancing an understanding of the physics of 2D and quasi-2D materials, the realization that with respect to electronic coupling and ultrafast carrier behavior, 2D behavior is already found in 3D crystals, enables the design of optoelectronic devices that take full advantage of some of the novel properties of 2D materials without requiring the more involved fabrication associated with genuine few-layer structures. Our central finding may thus greatly facilitate design and implementation of quasi-2D devices where the desired band-gap engineering might be fulfilled, but the crucial knowledge about the electronic coupling and ultrafast carrier dynamics, is unknown.

In order to make these points more clearly in the manuscript, we situate our work more explicitly in this context in abstract (p. 3), introduction (p. 4) and discussion (p. 18).

2. Normal and spectator Auger features in fig 3: The authors measure the Auger spectra far from the resonance at a photo energy of 520 eV in order to determine the normal Auger component and they use it in order to disentangle normal and spectator component. In my opinion this procedure is not well explained in the article. I suggest to the authors to move the part “Normal and spectator Auger fits” in the main manuscript. This would make the article more clear for people who are not in the field.

We thank the reviewer for the suggestion to move the determination of spectral fits into the main manuscript from SI and we agree this knowledge is crucial to ultimately determine charge delocalization times. In order to not distract from the main narrative we have left the detailed description of spectral fits in the SI, but inserted some additional clarifications and delineated the procedure for how we obtained the normal Auger fit in the main text.

p. 8: “[... see Supplementary Information for details). Briefly, the normal Auger features for both Sn $M_{4,5}N_{4,5}N_{4,5}$ transitions correspond to final state with d^8 atomic configuration (electronic configurations 1S , 1G , 3P , 1D and 3F). We fit transitions to each of these final states by incorporating state-dependent Coulombic repulsion and a screening term in the two-hole final states, giving excellent agreement with theory³⁴. The resulting fit parameters for the normal Auger spectra were then also used to capture the spectator Auger features after adding a spectator shift to account for the increased screening by the extra electron located in the conduction band of SnS₂ in resonant Auger spectroscopy.”

3. Time scale determination: The authors “expect to observe competing processes on a time scale of about 170 as to 17 fs”. It is not clear to me how do they estimate this interval.

Charge delocalization times are computed directly from intensity ratios (eqn. (1) in manuscript), and their accuracy is limited predominantly by noise associated with counting electrons. We use a core-hole lifetime $\tau_{c-h} = 1/k_{c-h} = 1.69$ fs and a conservative limit requiring that intensities be no more than one order of magnitude apart to still enable determination of delocalization times, i.e. $0.1\tau_{c-h} \leq \tau_{deloc} \leq 10\tau_{c-h}$. In order to clarify this point, we added on p. 13:

p. 13: “[... on this edge.] The experimentally accessible charge delocalization time-scale is determined from the accuracy with which RPES intensities can be determined, requiring that intensities be no more than one order of magnitude apart to still enable determination of delocalization times, and hence $0.1\tau_{c-h} \leq \tau_{deloc} \leq 10\tau_{c-h}$.”

4. In the last part of the text, the authors talk about interlayer and intralayer hopping time. Is the hopping time equivalent to the delocalization time estimated by core-hole clock spectroscopy? In the band picture the carrier can hop from one atom to the neighbor one in a characteristic time (hopping time) $t = \hbar/W$ where W is the width of the conduction band. What is width W of the conduction band estimated by DFT calculations reported in Fig 4? Is the estimated value of W compatible with the delocalization time measured by the experiments?

The reviewer is correct that we use various terms interchangeably. We have remedied this issue in the revised manuscript and use now uniquely the term “delocalization time” throughout.

These times can in principle indeed be compared to characteristic hopping times estimated e.g. from DFT calculations ($\tau_{hop} = \hbar/W$ with W the bandwidth): This yields an intralayer hopping time of ~300 as and an interlayer delocalization time in excess of many fs (see dispersion in Figure 4 and Supplementary Figure 3). Despite the remarkable compatibility with our RPES measurements, we emphasize that DFT calculations neglect the presence of the core-hole. Given the well-known issues if DFT calculations in obtaining correct bandwidths, *the most important conclusion from these estimates is the recognition that direct dynamical measurements such as core-hole-clock spectroscopy are necessary to obtain reliable insight into the carrier delocalization dynamics.*

In order to clarify this point in the manuscript, we have added the following on p. 17:

p. 17: “A comparison of these delocalization times with a simple estimate of hopping times obtained from the DFT band structure calculations (Fig. 4 and Supp. Fig. 3, $\tau_{hop} = \hbar/W$ with W the bandwidth) shows that such estimates (~ 300 as intralayer hopping and $\gg 1$ fs interlayer hopping) are compatible with our experimentally determined delocalization times. Note however that hopping times and delocalization times are not expected to be identical due to the presence of the core hole in the RPES measurements and possible difficulties of DFT calculations to capture bandwidths correctly.”

5. Inter and intralayer tau delocalization: Can the authors compare the values for intra and interlayer delocalization in SnS₂ with the delocalization time theoretically estimated or experimentally measured on other semiconductors or layered materials such TMDs and graphene? Is the carrier delocalization mechanism measured in SnS₂ particularly fast with respect to other materials?

Our approach uniquely affords a measure of intra- and interlayer delocalization times, and no *direct* comparisons are presently available. Transient absorption spectroscopy in layered materials such as MoS₂, WS₂ and WSe₂ has primarily probed excitonic species, vastly different from the carrier delocalization events accessible to core-hole-clock spectroscopy. Perhaps the closest estimate of interlayer charge-transfer times comes from a recent study of ultrafast charge-transfer in thin MoS₂/WS₂ heterostructures (Hong et al., *Nat. Nanotech.* **9**, 682-686 (2014)) which found an interlayer charge-transfer time of <50 fs. While this is compatible with our finding between layers of SnS₂ (interlayer delocalization time >2.7 fs), neither measurement is at present sufficient to comment on differences among layered materials. We reiterate that core-hole clock spectroscopy provides a unique experimental platform to directly compare intralayer vs. interlayer delocalization times that is critical in understanding 2D layered materials, and further studies will be necessary to compare different materials.

We now discuss this point on p. 18:

p. 18: “Our measurements constitute the first observation of anisotropic carrier dynamics in van der Waals layered materials, complementing efforts to tailor and understand the band structure of these materials e.g. by ARPES^{13,14,15,16}. While the extent to which layer-decoupling varies across this class of materials is at present unknown, we anticipate that our approach enables investigations into variations in interlayer coupling in response to stacking symmetry, twist angle and formation of heterostructures, complementing steady-state spectroscopies such as ARPES and photoluminescence³⁹. Our findings further strongly suggest the possibility to create long-lived interlayer excitons⁴⁰ in few-layer and even in bulk crystals, since interlayer delocalization between neighboring SnS₂ sheets is at least a factor of 10

slower than intralayer delocalization. This process may be quite fast in the presence of an energetic gradient, as indicated by a recent report of interlayer charge-transfer times of < 50 fs in MoS₂/WS₂ heterostructures⁴¹. Note that this time-scale is fully consistent with our measured interlayer delocalization times.”

Reviewer #2

1. It is not clear why the hole delocalization time for B_{1,2} is so long. While it is argued that the presence of p_z character means this is associated with interlayer coupling, it does not seem obvious that it is exclusively (or even predominately) interlayer in nature. Looking from the orbital perspective, table I shows significant (30~60%) p_{x,y} character. Presumably, this is also intralayer in nature and so while one may not expect the electron to delocalize quickly in z, why does it not delocalize in-plane? Could this instead be tied to a lack of any significant Sn (S) character in the band?

We thank the reviewer for their keen observation of orbital compositions involved in intralayer and interlayer dynamics. Several points must however be considered:

- a) The reviewer focuses on the orbital composition at Γ ; core-hole-clock spectroscopy does however not distinguish between different points in the Brillouin zone, and as a result an average distribution of orbital contributions at multiple points in the Brillouin zone should be considered (see Table 1). Thus, p_{x,y} and p_z orbitals contribute approximately equally to B_{1,2}.
- b) The reviewer is correct that bands A_{1,2} and B_{1,2} differ significantly in their Sn 5s character. On its own, this is however not sufficient to understand the different carrier dynamics: We are exciting from Sn 3d levels, and by electric dipole selection rules can only access Sn p (and f) orbitals. The (weak) excitation of A_{1,2}, formally forbidden by selection rules, is only possible if Sn 5s is strongly hybridized with S orbitals with p-character, i.e. in-plane 3p_{x,y} (Table 1). This strong hybridization is supported by the delocalization times for band A_{1,2}.
- c) In contrast, B_{1,2} (primarily Sn and S, p_{x,y} and p_z character, Table 1) is accessible by dipole rules without a need for strong hybridization. While this suggests the possibility for simultaneous intra- and interlayer delocalization, Sn 5p_{x,y} and S 3p_{x,y} can be expected to exhibit weaker overlap and hence hybridization than Sn 5p_z and S 3p_z orbitals, given the geometry of the unit cell. Hence, while both Sn and S p_{x,y} and p_z orbitals in B_{1,2} may be excited, we expect out-of-plane p_z orbitals and hence interlayer delocalization to dominate the dynamics in this region.

In order to make this point more clearly, we have added the following on p. 15:

p. 15: “[... in Table 1.] Since RPES cannot distinguish between different points in the Brillouin zone, small variations in the overall similar orbital composition at

both Γ and M should be averaged. Regions $B_{1,2}$ differ markedly from $A_{1,2}$ by containing significant out-of-plane orbital character from Sn and S p_z contributions and by lacking Sn 5s character. As reflected in the weak spectral intensity, excitation of regions $A_{1,2}$ is by dipole selection rules only allowed due to strong hybridization with S $3p_{x,y}$ orbitals. $A_{1,2}$ is thus primarily composed of strongly hybridized in-plane molecular orbitals. $B_{1,2}$ contains primarily Sn and S $p_{x,y}$ and p_z character. Of these, the Sn and S p_z orbitals are more strongly hybridized due to the atomic arrangement in the unit cell (see insert Fig. 1), and $B_{1,2}$ contains significant out-of-plane character.”

2. Along a similar line, the dispersion of the bandstructure in the in-layer vs out-of-layer directions may be more convincing than relying on the orbital character. The bandstructure shown seems to be only for in-plane high symmetry points (for a hexagonal supercell?). It would be nice if we could also see Gamma-A-Gamma, so we could see how the effective mass of A and B in the parallel and perpendicular directions compare.

We agree that the out-of-plane band structure from Γ -A offers a useful perspective. We added a new figure to the SI, illustrating the lack of significant dispersion in both valence and conduction bands along this direction (p. 31-32) and added the following to the manuscript:

p. 14: “[... Information).] The full band structure including Γ to A is shown in the Supplementary Information (Supp. Fig. 3).”

p. 18: “[... transfer dynamics.] This is also fully consistent with the band structure, which shows negligible dispersion along the Γ to A direction (see Supplementary Information).”

3. The manuscript states that a “small I_{SA} fraction reflects fast charge delocalization (high k_{deloc})”, and vice versa. This seems to directly contradict equation (1), $k_{deloc} = k_{c-h} I_{SA}/I_{NA}$. From my understanding, the localized electron screens the core-hole more effectively, so eqn. (1) seems incorrect. Should it instead be I_{NA}/I_{SA} ?

Yes, the reviewer is absolutely correct, this is a typo. The equation has been modified to reflect the correct equation: $k_{deloc} = k_{c-h} \frac{I_{NA}}{I_{SA}}$.

REVIEWERS' COMMENTS:

Reviewer #1 (Remarks to the Author):

In my original report, I put in doubts the suitability of the paper for Nature Communications, because I was not totally convinced that the impact of these results on the physics of 2D materials was high enough to warrant the publication on this journal. In the revised version of the paper, the authors put more emphasis on the novelty of their technique and to the fact that they find an evidence that the ultrafast carrier dynamics of layered bulk material (in-plane and out-of-plane) is strongly anisotropic. I have appreciated how the authors rewrote the abstract and the introduction to emphasize better these aspects. The authors have satisfactorily responded to all my questions and made the necessary changes to the manuscript.

I hope that core-hole-clock spectroscopy technique could be used soon to study the same phenomena and to prove the dynamical anisotropic behavior of carriers in other layered materials like Mo- and W-based transition metal dichalcogenides which have been already largely explored by the 2D materials' community.

I recommend therefore this revised version of the paper for publication in Nature Communications.

Reviewer #2 (Remarks to the Author):

The authors have adequately addressed all of my concerns. I can now recommend for publication.